

# Dry versus wet? Implications on aerosol impaction and organic volume fraction

Hansol D. Lee[1], Chathuri P. Kaluarachchi[1], Elias S. Hasenecz[1], Zhehao Zhu[1], Eduard Popa[1], Elizabeth A. Stone[1], and Alexei V. Tivanski[1]

[1] Department of Chemistry, University of Iowa, Iowa City, IA 52242, USA.

*Correspondence to*: Alexei V. Tivanski (alexei-tivanski@uiowa.edu)

**Abstract.** Understanding the impact of sea spray aerosols (SSA) on the climate and atmosphere requires quantitative knowledge of their chemical composition and mixing states. Furthermore, single particle measurements are needed to accurately represent large particle-to-particle variability. To quantify the mixing state, organic volume fraction (OVF), defined

as the relative organic volume with respect to the total particle volume, is measured after generating and collecting aerosol particles, often using deposition impactors. In this process, the aerosol streams are either dried or kept wet prior to impacting on solid substrates. However, the atmospheric community has yet to establish how dry versus wet aerosol deposition influences the impacted particle morphologies and mixing states. Here, we apply complementary offline single particle atomic force microscopy (AFM) and bulk ensemble high performance liquid chromatography (HPLC) techniques to assess the effects of

dry and wet deposition modes on the substrate-deposited aerosol particles' mixing states. Glucose and NaCl binary mixtures that form core-shell particle morphologies were studied as model systems, and the mixing states were quantified by measuring the OVF of individual particles using AFM and compared to the ensemble measured by HPLC. Dry deposited single particle OVF data positively deviated from the bulk HPLC data by up to 60%. The positive deviation was attributed to significant spreading of the NaCl core upon impaction with the solid substrate, which is not readily evident in AFM imaging and leads to

underestimation of the core volume. NaCl core spreading under impaction was confirmed by imaging dry deposited NaCl particles. This problem was circumvented by a) performing wet deposition and thus bypassing the effects of the solid core spreading upon impaction and b) performing a hydration-dehydration cycle on dry deposited particles to restructure the deformed NaCl core. Both approaches produced single particle OVF values that converge well with the bulk and expected OVF values, validating the methodology. These findings illustrate the importance of awareness in how conventional particle

deposition methods may significantly alter the impacted particle morphologies and their mixing states. Our work can help improve quantification and predictions of chemical mixing states of atmospherically-relevant aerosols.





# 1 Introduction

The chemical composition of the ocean and sea surface microlayer (SSML) directly affects the mixing states of sea spray aerosol (SSA), which is generated by film and jet drops. (Prather et al., 2013;Jacobson, 2001;Vignati et al., 2010;de Leeuw et al., 2011;Wang et al., 2017) Typically existing as submicrometer-sized aerosols with inorganic core encased by organic shell (core-shell), their heterogeneous mixing states derive directly from the complex variety of organic, inorganic, and biological species in the SSML and ocean water. (Cochran et al., 2017;Ault et al., 2013) Nascent SSAs affect the Earth's climate and atmosphere through radiative forcing, directly by scattering and absorbing solar radiation, and indirectly by acting as cloud condensation or ice nuclei (Lee et al., 2017b;Haywood and Boucher, 2000;Jacobson, 2001); however, their chemical and biological complexity that control the mixing state, hinder our ability to accurately predict their climate cooling abilities. (Lee et al., 2017b;Lee et al., 2017a)

Thus, with relatively recent single particle methodology developments, characterizing the SSA-relevant aerosol chemical compositions and ensuing physicochemical properties can be performed through offline bulk ensemble and/or single particle techniques. (Laskina et al., 2015;Ault et al., 2013;Cochran et al., 2017;Estillore et al., 2017;Haywood and Boucher, 2000;Prather et al., 2013;Cochran et al., 2016;Morris et al., 2016;Fuentes et al., 2010;King et al., 2012;Schill et al., 2015;Morris et al., 2015;Schill and Tolbert, 2014;Jacobson, 2001;Collins et al., 2014;Guzman et al., 2012) Quantitative bulk ensemble techniques sample large particle numbers; however, they cannot provide individual particle specificity, which is crucial when studying submicrometer-sized aerosol particles that display large particle-to-particle variability. For example, single particle level studies are well-suited to study ice nucleation, because only 1 out of 106 particles form ice clouds that cool the Earth. (DeMott et al., 2017) Therefore, single particle techniques such as electron microscopy or atomic force microscopy (AFM) are highly attractive, because they offer imaging capabilities with nanometer-level spatial resolution on substrate-deposited individual particles. Additionally, AFM measurements can be performed under controlled relative humidity (RH), while producing high-resolution 3D height and phase images, and directly measuring viscoelastic properties of materials. (Lee et al., 2017a; Lee et al., 2017b) In this regard, simultaneous acquisition of the 3D height and phase images over individual particles can be used to quantify their mixing states, or organic volume fraction (OVF). OVF measurements require quantification of phase-separated organic component and the total particle volumes, which can be determined for individual particles using Eq. (1),

$$OVF_{particle} = \frac{V_{org}}{V_{total}} = \frac{V_{total} - V_{inorg}}{V_{total}},$$  (1)

where $OVF_{particle}$ is individual particle organic volume fraction, and $V_{total}$, $V_{org}$, $V_{inorg}$ are the volumes of the total particle, organic shell, and inorganic core, respectively. Previously, similar methodology was used without systematic validation, to





report upper-limit approximations of $OVF_{particle}$ for mixtures of NaCl, glucose, and laminarin. (Estillore et al., 2017) In this work, the goal was to provide a quantitative validation of the methodology.

SSAs are typically collected on substrates via laboratory generation or during a field study, then are either dried to a relatively low RH (ca 20-35%, dry deposition) or kept wet at relatively high RH (> 75%, wet deposition) prior to being sent to an impactor for substrate deposition. (Ovadnevaite et al., 2017;Facchini et al., 1999;Lee et al., 2017a) Despite the extensive use of both methods for single particle analysis, the aerosol community has yet to quantify the effect of dry or wet deposition on experimentally determined particle morphologies and ensuing physicochemical properties. On one hand, wet deposition may cause splattering of particles, in which organic volume is lost and thus underestimated. On the other hand, dry deposition may cause physical deformation of particles upon impaction on a hard substrate, possibly introducing a source of error.

Here, we address two questions, what are the effects of dry and wet deposition on atmospherically relevant, phase-separated aerosol particle morphologies? And how does this affect the assessment of mixing state by single particle OVF measurements? To answer these questions, we chose glucose and sodium chloride (NaCl) as a model of a core-shell particle, at two molar ratios (ca 1:2 M and 1:8 M) and applied single particle AFM and bulk ensemble liquid chromatography techniques. Glucose is a good model system because of available data on the relationship between RH and viscosity, as well as its ability to access solid, semisolid, and liquid phase states at subsaturated RH. (Lee et al., 2017b;Song et al., 2016) NaCl and glucose mixtures also produce core shell particles, with solid-liquid phase separation evident in AFM phase imaging. Both chemical systems are highly relevant to the SSA. (Jayarathne et al., 2016) Further, both glucose and NaCl are surface inactive species, thus OVF is not expected to be size-dependent. (Cochran et al., 2017) In the following sections, we begin by introducing the observed core-shell morphology of phase-separated binary component particles using AFM. Next, quantified OVF results for 1:2 (M) and 1:8 (M) glucose:NaCl is discussed under dry and wet deposition conditions. Finally, an experimental approach to restructure the dry deposited particles by performing a hydration and dehydration cycle is introduced and validated.

## 2 Experimental

### 2.1 Particle generation

Glucose and sodium chloride (NaCl) were purchased from TCI and Fisher Chemical, with 98% and 99% purity, respectively. Both were dissolved without additional purification in ultrapure water (>18 MΩ·cm), to generate 0.1 M glucose solutions for mixtures and 0.1 M NaCl solutions for pure NaCl. From each solution, corresponding particles were generated using a custom-made bubbler system with ½" corrugated stainless steel or Teflon tubing and Swage connections. (Cochran et al., 2016) For dry deposition, the particle stream was sent to two diffusion dryers used to maintain approximately 20% RH (70 cm length, created in house), and to a mixing chamber with 26 L/min clean air by-pass. For wet deposition, the diffusion dryers were removed and the 26 L/min dry air bypass was sent through ultrapure water to achieve ~80% RH in the mixing chamber.



## 2.2 Sample collection

Particles were substrate-deposited using a Micro Orifice Uniform Deposit Impactor (MOUDI) (MSP, Inc., Model 110) on hydrophobically treated silicon wafers (Ted Pella, Inc.).(Lee et al., 2017a) The substrate was cleaned with ethanol and high purity N2 gas (99.998%) prior to use. High Performance Liquid Chromatography (HPLC) samples were collected on 47 mm

Teflon filters (PALL Life Sciences). Prior to sample collection, field blanks were obtained before each experiment (total 3).

## 2.3 AFM imaging

Molecular force probe 3D AFM (Asylum Research, Santa Barbara, CA) was used for particle imaging at ambient temperature (20 - 25 °C) and pressure. A custom-made humidity cell was used to control the RH (~3% - 97%). (Lee et al., 2017a) Silicon nitride AFM tips (MikroMasch, Model NSC35) with nominal spring constant range of 5 – 16 N/m were used to image particles.

Prior to imaging, the humidity cell RH value of 25-30% was maintained for at least 10 minutes, to allow the particles to reach equilibrium with the surrounding humid air in the cell. AC mode was used to image individual particles and collect information on their 3D height and phase. Igor Pro particle analysis tool was used to calculate the core and total particle volume (See SI for details). The single particle data of organic volume fraction and volume equivalent diameter are both shown with a mean value and error bar, which represents two standard deviations.

## 2.4 Bulk ensemble measurements

Teflon substrates were extracted by 10 minutes mechanical stirring, 30 minutes sonication, and an additional 10 minutes of mechanical stirring into 5.00 mL ultrapure water and extracts were filtered via a 0.45 µm polypropylene Whatman filter and analyzed via HPLC. (Jayarathne et al., 2016;Rathnayake et al., 2016) Glucose content was quantified using high performance anion exchange chromatography (ICS-5000, Dionex) with pulsed amperometric detection using a Dionex MA1 guard and

analytical columns. Sodium and chloride content were quantified using HPLC with conductivity detection as previously described. (Jayarathne et al., 2016) The bulk organic volume fraction data, which was converted from mass-based HPLC measurements to volume using the densities of glucose and sodium chloride, is shown with a mean value and error bar, which represents propagated analytical uncertainty (See SI). We note that the mention of bulk measurements in the text is solely dedicated to describing the measurements performed using HPLC.

## 3 Results & Discussion

### 3.1 Morphology of phase-separated binary component particles

After generation of glucose and NaCl mixture particles by dry and wet deposition methods outlined above, AFM imaging was used to obtain the 3D height and phase images and validate a "core-shell" morphology for two different molar fractions (Fig. 1). Images do not show changes of morphology from core-shell, but instead the core is more evident under wet deposition.



Here, RH was maintained at a constant range of 25 – 35% throughout the imaging experiment, to minimize "phase bleeding" and water uptake (Fig. S1). In this work, phase bleeding describes an instance in which the viscosity of the two phase-separated materials is too similar at a given RH, and therefore the phase contrast between organic and inorganic components is relatively weak in the AFM images. A high phase contrast image aids in correctly differentiating the core and shell phase boundaries.

For this particular system, a dramatic increase in phase bleeding is evident below 25% RH, due to a significant increase in the glucose viscosity as it becomes more solid-like, which closely resembles the NaCl viscosity at this RH value (Fig. S1). (Song et al., 2016) Since phase imaging inherently relies on measuring differences in tip-sample interactions originating from different viscoelastic properties, lowering the RH will further converge the two different viscosities of organic and inorganic components together, lessening the accuracy of the core and shell phase boundary determination. Although higher RH values

would produce less "phase bleeding", significant particle water uptake will also introduce erroneous OVF values, due to an increase in particle volume from water uptake. For chemical systems with a discrete deliquescence point, such as ~75% RH for NaCl, the upper limit RH value is clear. However, for chemical systems that continually uptake water, such as glucose, the lack of discrete deliquescence point complicates the matter. (Lee et al., 2017a) The value of 25% RH was thus used for the measurements of OVF, because growth in size from water uptake was measured to be less than 1.03, or merely 3% growth

while showing excellent phase contrast.(Lee et al., 2017a) Other chemical systems may also have optimum RH ranges that minimize water uptake, while allowing for clear distinction of the core and shell. Further discussion of the environmental conditions for phase imaging is provided in the supporting information.

### 3.2 OVF measurements on 1:8 (M) glucose:NaCl

From Figure 1, a phase imaging mask of the area was used to identify the boundary of the inorganic core and the organic shell,

which was used to quantify the inorganic core and the total particle volumes, the difference of which is organic volume (masks shown in Fig. S2). The area mask is then transferred onto the 3D image, and the OVF is quantified using Equation 1. We note that this methodology assumes that a) the areas of the top and bottom of the core, are nearly identical and that b) no organic material is present below the core. Below, we will show that the first assumption fails, under dry deposition conditions.

Figure 2A shows OVF versus volume-equivalent diameter data on individual 1:8 (M) glucose:NaCl mixture particles from dry and wet deposition modes. The dotted black line is the expected OVF value. Volume-equivalent diameter was quantified by relating volume of hemisphere to the diameter. (Morris et al., 2016) Bulk OVF ($OVF_{bulk}$) was calculated from the masses of glucose and NaCl, determined by their respective densities (Equation S1). More information on the expected OVF, calculated from the measured organic and inorganic mole fractions from the bulk solution measurements, can be found in the supporting

information (Equation S5). The grey-shaded area represents ±10% deviation from the mean as a reference. In agreement with our expectations, the bulk OVF data does not show any size dependence and instead shows good overlap with the expected values. Also, the bulk data shown here were wet deposited, but no significant differences between dry and wet deposition were observed in $OVF_{bulk}$ (Fig. S3). For this mixture, single particle AFM OVF of dry deposited particles show a significant,





positive deviation away from the expected and bulk OVF values. In fact, the deviation is nearly 60% higher than the expected value. But, there is no clearly evident size dependence on measured OVF. Similarly, wet deposited particles also do not show size dependence, but instead show excellent overlap with the expected and bulk OVF values. Moreover, contrary to our concerns outlined in the beginning, the wet-deposited particles were individual with well-resolved core-shell morphology

without any evidence of spattering.

To better illustrate the differences in measured OVF means and variance from dry and wet deposited particles, statistical comparisons of sample means were performed from constructed histograms (Fig. 2B). We note the histograms were obtained by combining individual particle data from all size ranges as no clear size dependence was observed. The probability

distribution functions are shown as the colored solid lines. The distribution of OVF data from dry deposited particles showed mean and standard deviation values of $0.54 \pm 0.06$ (n = 222). Wet deposited particles yielded mean and standard deviation values of $0.34 \pm 0.05$ (n = 134). As a reference, the expected OVF and standard deviation is $0.37 \pm 0.03$, constructed assuming 10% deviation from the mean. Overall, the histograms show that wet deposited particle data overlapped well with the expected and bulk values, in sharp contrast to dry deposited particles that yielded nearly 60% deviation. The deviation of OVF is likely

due to significant spreading of the core, but also contributed by the 3% contribution of water in the organic volume calculation and potential organic layer that is hidden underneath the core. Below, we further elucidate the deviation using statistical tools.

To reject the null hypothesis that dry deposited versus wet deposited OVF data are statistically insignificant, an independent samples t-test (dark orange vs. blue) was performed. This comparison will be used to calculate the probability of obtaining the

observed differences from dry versus wet OVF values out of random chance. Also, to quantify how the sample means from dry versus wet deposited OVF data compared to the expected value, Cohen's d was calculated, which represents how far away the sample mean is with respect to the expected value, but normalized to the distributions shown. Equations for t-test and Cohen's d are in the supporting information (Eq. S6 & S7). First, comparing the dry- versus wet-deposited OVF distributions, two tailed t-test confirmed extremely strong statistical significance, $t(354) = 32.838$, $p < 0.0001$. Thus, statistical analysis

shows that there is less than 0.01% probability of random chance that can explain the differences in the sample means. Therefore, t-test confirmed that dry and wet deposited particles produced statistically different results, which allows us to reject the null hypothesis outlined in the beginning. Also, Cohen's d calculations for the dry (dark orange) and wet (blue) OVF data versus the expected value was 4.142 and -0.419, respectively. Therefore, the wet deposited particles were statistically much closer to the expected value then the dry deposited particles.


To briefly recap, OVF data from dry deposited 1: 8 (M) glucose and NaCl particles showed strong deviations, whereas the wet deposited particles showed excellent overlap with the predicted OVF values. In the beginning, we introduced the drawbacks of performing dry and wet deposition methods. For dry deposition, the particle morphology may deform due to a violent impaction with the substrate, which may introduce errors when performing OVF measurements that rely on assumptions on



the shape of the core. If so, the OVF overestimation, or analogously core volume underestimation from single particle analysis, can be explained by the core spreading upon impaction. Under dry deposition conditions, the solid NaCl core encased by glucose can spread underneath the glucose shell from the impaction, which is not detected in AFM phase imaging that probes the very surface of the particle. This effect is expected to produce an artificially higher OVF. Below, we further investigate

this phenomenon.

### 3.3 Evidence of dry deposited NaCl spreading

To determine the extent of NaCl spreading upon impaction, pure NaCl was dry deposited under the same conditions as the previous glucose and NaCl mixture. Figure 3A shows an AFM 3D height image of dry deposited NaCl. The transition from purple, red, to green indicates a gradient of height. The morphology-deforming effects of the dry deposition and impaction are

evident, where the top of the particle shows round, not sharp edges. The bottom of the particle has a significantly greater area than the top of the particle. This is more evident in Figure 3B, where cross-section of the particle is shown (denoted by red dashed line). For illustrative purposes, the expected cross-sectional area of NaCl, assuming no spreading, is shown with green striped area. In comparison to when there is no spreading, which should produce top and bottom areas that are identical in distance, axis shows direction-dependent spreading of nearly 300 nm that would not be considered when it is encased in organic

matter, and OVF is quantified. To validate that the observed data is attributed largely due to the NaCl particle spreading, Equation S9 was derived, which shows that at least 120 nm of the 300 nm observed spreading is directly due, not to the tip-shape convolution, but to the particle spreading (see SI). Once this was confirmed, the effectiveness of wet deposition was tested on a different mixture ratio of glucose and NaCl.

### 3.4 OVF measurements on 1:2 (M) glucose:NaCl

Figure 4A shows OVF versus volume-equivalent diameter data on individual 1:2 (M) glucose:NaCl mixture particles deposited dry and wet. In comparison to the bulk results, AFM OVF data from dry deposited particles show size dependence at approximately 250 nm in volume-equivalent diameter. To better illustrate this, we represent the sample set of dry deposited particles below this arbitrary size as a lighter shade of orange. Also, positive deviation away from the expected and bulk OVF values is observed with increasing particle size. Since bulk OVF results do not display any size dependence, the apparent size

dependence of AFM OVF values for dry deposited particles underscores the complex nature of how the morphology of solid particles changes as a result of impaction, thus introducing artificial dependence that prevents accurate reflection of mixing states of air suspended particles. Contrary to dry deposited particles however, AFM OVF data from wet deposited particles do not show size dependence, with excellent overlap with the expected and bulk OVF values. Moreover, similar to the 1:8 (M) mixture, the particles were also individual and did not show any evidence of spattering.

To better illustrate the differences in measured AFM OVF sample means and variance from dry and wet deposited particles, histograms were constructed (Fig. 4B). The OVF data distribution from dry deposited particles showed mean and standard deviation values of $0.64 \pm 0.06$ (n = 84) for particles less than 250 nm in size, and $0.82 \pm 0.04$ (n = 99) for particles greater



than 250 nm in size. Wet deposited particles showed mean and standard deviation values of 0.71 ± 0.07 (n = 146). As a reference, the expected OVF and standard deviation is 0.71 ± 0.06, again constructed assuming 10% deviation from the mean. Overall, the histograms show that wet deposited particle data overlapped much better with the expected and bulk values, in comparison to dry deposited particles. To confirm this statistically, same student's t test methodology was employed (see SI).

Analysis showed that the OVF results for wet deposited particles were statistically much closer to the expected value than the dry deposited particles.

### 3.5 Restructuring dry deposited particles by hydration-dehydration cycle

In certain situations, wet deposition may not be feasible. Therefore, an experimental approach was developed to restructure the previously dry deposited particles through water uptake (Fig. 5). Figure 5A shows two phase images of the same dry

deposited particle. The images were taken at 25% RH, prior to hydration-dehydration cycle (hyd-deh cycle) on the left and after the cycle on the right. The color schemes of a green organic shell and violet/red inorganic core remain the same as Figure 1. After dry deposition, the particles were initially kept at near 3% RH, until imaging at 25% RH. Then, the humidity was increased to > 80% RH which resulted in deliquescence of the particle and phase transition to liquid phase, confirmed by observing the drastic droplet size increase. (Morris et al., 2016) The RH was then slowly decreased back to 25% RH over ca

10 minutes, resulting in the dehydration of the particle. Data show that after the hyd-deh cycle, more of the inorganic core is clearly evident in AFM phase imaging, which can now be taken into account in volume calculations.

From the quantified OVF before and after the hyd-deh cycle, histograms were constructed (Fig. 5B). The distribution of OVF data before the cycle showed mean and standard deviation values of 0.51 ± 0.02 (n = 28). Particles after the cycle showed mean and standard deviation values of 0.38 ± 0.07 (n = 40). The expected OVF and standard deviation is 0.37 ± 0.03. Thus,

the hyd-deh cycle effectively lowered the OVF value of the same particle to be closer to that of the expected value. To confirm this statistically, same student's t test methodology was employed (see SI). Analysis showed that the particles after the cycle were within 99.9% confidence interval of the expected value, whereas the particles before the cycle were outside of the range. Furthermore, this experiment confirms the original hypothesis that the core spreading upon dry impaction produces erroneous OVF values, which can be remedied either by performing wet deposition, or dry deposition with subsequent hyd-deh cycle.

However, we note that this methodology relies on water uptake and phase transition from solid into liquid and then back to solid state. At the moment, this approach cannot be readily used on systems that do not undergo solid to liquid phase transitions.

### 4 Conclusions

In summary, offline single particle AFM and bulk ensemble HPLC techniques were used to assess dry and wet deposition effects on the generated and impacted aerosol particles' mixing states that display core-shell morphology. It was tested by

quantifying the organic volume fraction of model binary component mixture particles composed of glucose and NaCl at two different molar ratios. For both 1:2 (M) and 1:8 (M) glucose:NaCl mixtures studied here, the measured single particle OVF





from dry deposition showed positive deviations away from the expected values. In the case of 1:2 (M) glucose:NaCl particles, even an artificial OVF size dependence was observed. The positive deviation was found to originate from the inorganic core spreading, resulting in overestimation the volume of the core. However, performing wet deposition on the same mixtures produced single particle OVF values that converged well with the expected values, with extremely strong statistical

significance. Furthermore, data showed that performing a hydration-dehydration cycle on previously dry deposited particles effectively forced core restructuring through solid to liquid to solid phase transitions, also reproducing accurate single particle OVF measurements. This particular approach is useful for studies that cannot perform wet deposition, but for which the accurate particle morphology is still required. In the future, an analytical study to further test the dry versus wet deposition methods will be performed on increasingly complex mixtures of organic and inorganic systems, including surface-active

species. Ultimately, these findings bring us closer toward improving the accuracy of climate predicting models that heavily rely on knowledge of the chemical composition and resultant mixing states of atmospherically-relevant aerosol particles.

**Author contribution**

Project administration – EAS and AVT. Experiments – HDL, CPK, ESH, and EP. Data analysis – HDL, CPK, ESH, and ZZ. Writing – HDL, ESH, EAS, and AVT.

**Competing interests**

Authors declare no competing interests.

**Acknowledgements**

This work was supported by the National Science Foundation (NSF) through the NSF Center for Aerosol Impacts on Chemistry of the Environment under grant no. CHE 1801971. Any opinions, findings, and conclusions or recommendations expressed in

this material are those of the authors and do not necessarily reflect the views of the National Science Foundation.

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

**Figures**

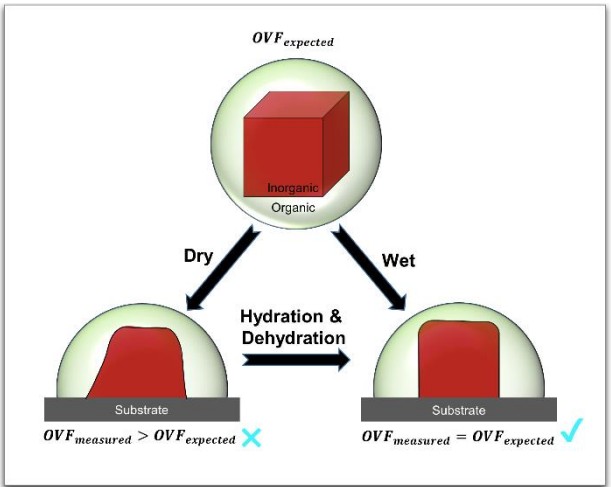

**TOC.**





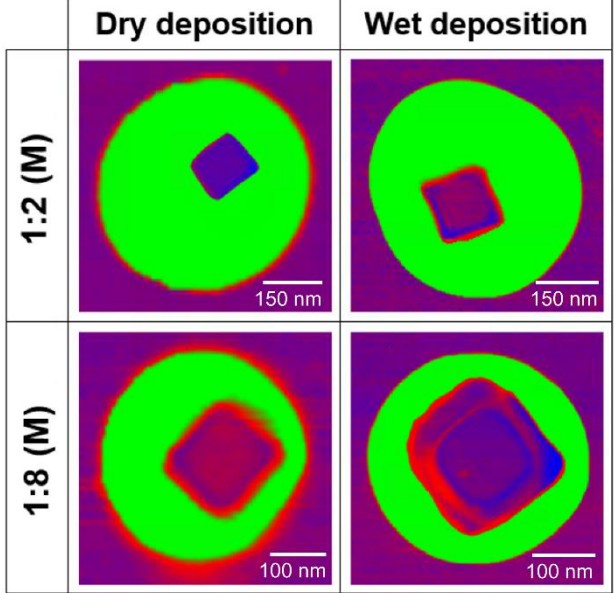

**Figure 1. AFM phase images of glucose:NaCl particles at 25 – 35% RH. The green and violet/red colors represent the organic and inorganic components of the particles, respectively.**



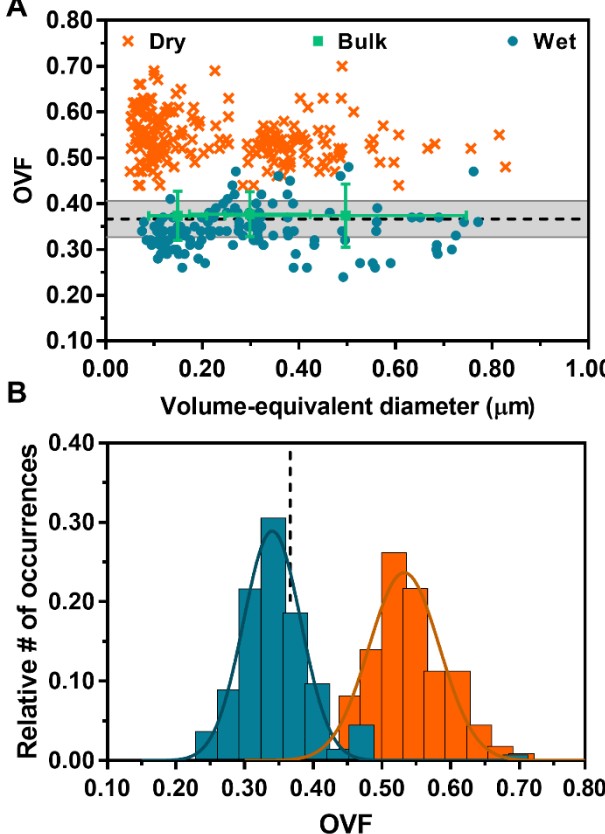

**Figure 2. AFM single particle OVF values (dry deposition: dark orange crosses, wet deposition: blue dots) versus volume-equivalent diameter for 1:8 (M) glucose:NaCl (A). The bulk OVF data (green squares) is shown with x-axis and y-axis error bars, obtained from the MOUDI size cut-off range and total experimental propagated error, respectively. Expected OVF value (black dotted line) is measured and calculated from bulk solution mixing ratios with known densities of solutes, assuming ±10% error (grey shaded region). AFM OVF histograms (same color legend schemes retained from above) collected over individual particles (B). Gaussian function fits are shown by the colored lines, yielding the most probable OVF values and corresponding standard deviations.**



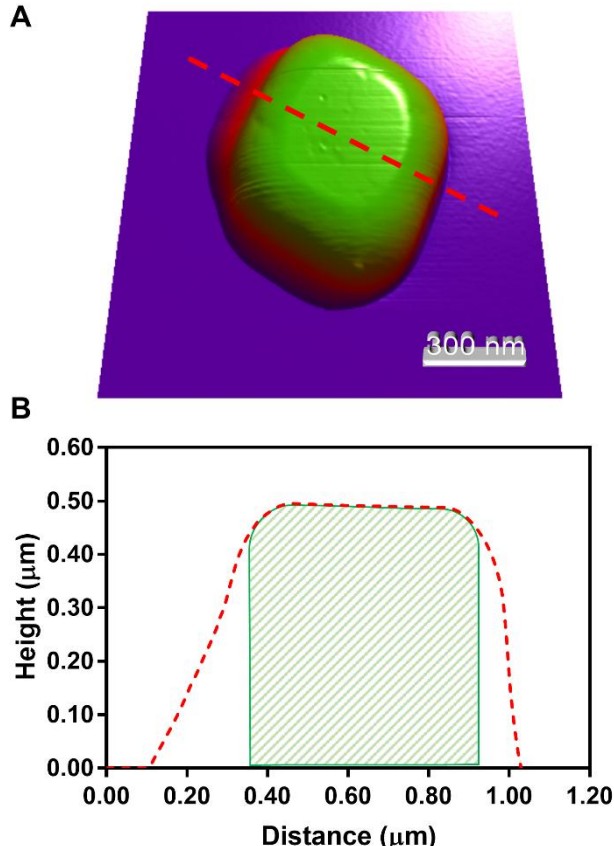

**Figure 3. AFM 3D height image of a dry deposited NaCl particle (A). The NaCl particle cross-section (B) that was measured (red dashed line) compared to the estimated, when assuming a cubic particle shape with no spreading (green striped region). The difference between the red and green lines corresponds to an underestimation of the NaCl core volume when significant spreading occurs.**



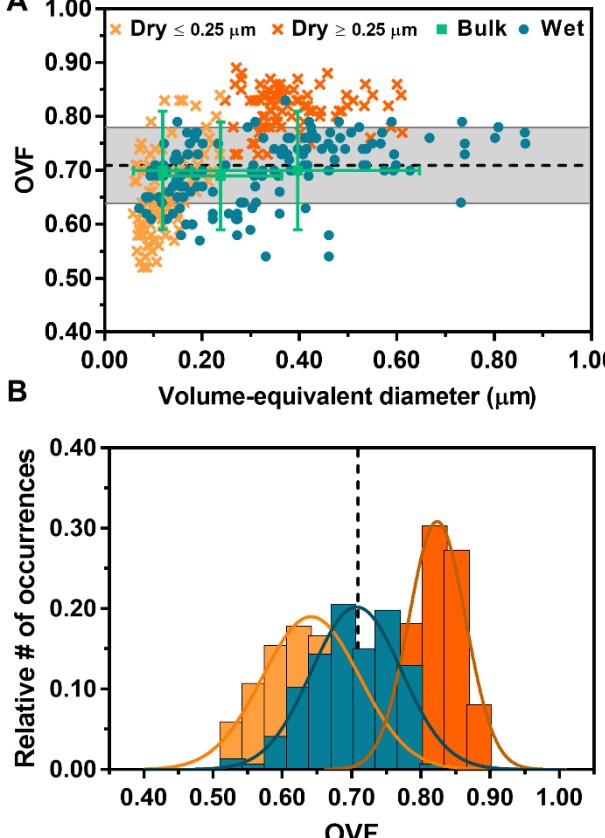

**Figure 4. AFM single particle OVF values (dry deposition: light and dark orange crosses, wet deposition: blue dots) versus volume-equivalent diameter for 1:2 (M) glucose:NaCl (A). The dry deposition is shown in two colors to distinguish the data set that exhibits apparent size dependence. The bulk OVF data (green squares) is shown with x-axis and y-axis error bars, obtained from the MOUDI size cut-off range and propagated analytical uncertainty, respectively. Expected OVF value (black dotted line) is measured and calculated with bulk solution mixing ratios with known densities of solutes, assuming ±10% error (grey shaded region). AFM OVF histograms (same color legend schemes retained from above) collected over individual particles (B). Gaussian function fits are shown by the colored lines, yielding the most probable OVF values and corresponding standard deviations.**



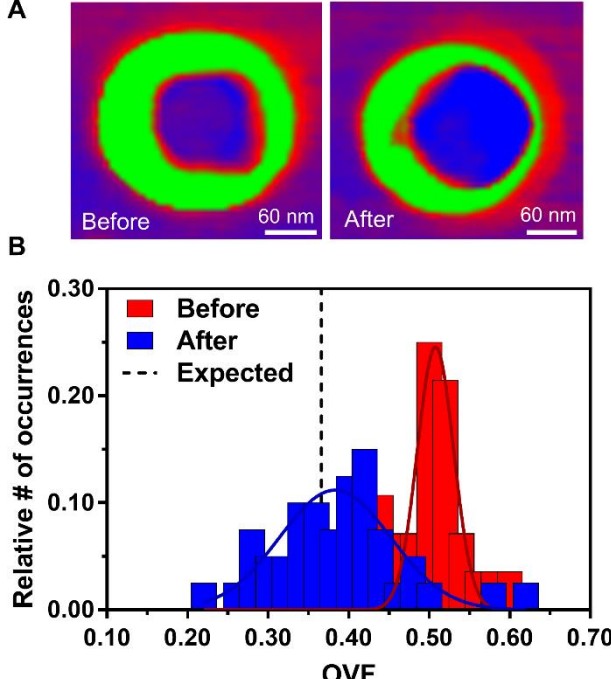

**Figure 5. AFM phase images of the same particle at 25% RH, before and after hydration-dehydration cycle (A). AFM OVF histograms for 1:8 (M) glucose:NaCl collected over individual particles (B). The red bars indicate particles before the cycle. The blue bars indicate the same particles after the cycle. Gaussian function fits are shown by the colored lines, yielding the most probable OVF values and corresponding standard deviations. The expected value is shown as the black dashed line.**