# Peer review of "Figure S1. AFM phase images of 1:8 (M) glucose:NaCl particle at varying RH (left - 18% RH, right - 36% RH). Image from 36% RH shows greater contrast between the core and shell, which aided in OVF quantification."

_Atmospheric Measurement Techniques, 2018_

## Referee Comment (RC1) · Anonymous Referee #1 · 13 Dec 2018

This paper investigated dry and wet deposition effects on the generated and impacted aerosol particles' mixing states that display core-shell morphology. In general, the techniques are well-used and the manuscript is well written. However, some suggestions are worth noting. 1 The abstract section is too long. Only key information is needed.

2 As we know, levoglucosan dominates the sugar particulates, while little glucose exists in the atmosphere. Meanwhile, more details about glucose (or particle phase sugar) should be added in the Introduction section.

3 Section 3.1: Authors employed the RH of 25% to measure OVF. However, RH in the real atmosphere is usually higher than 30%. Additionally, whether the authors considered the particle hygroscopicity?

4 For the equation S1, which kind of density is used here? Please explain in detail.

5 The authors used one paragraph to describe the t-test and null hypothesis. But t-test is a simple statistical tool and only the significant result should be put in manuscript.

6 Conclusion section. The atmospheric implications of the experimental results should be discussed.

---

## Referee Comment (RC2) · Anonymous Referee #2 · 18 Dec 2018

The authors describe the effect of dry and wet deposition on the mixing states of organic/NaCl particles using atomic force microscopy and bulk ensemble HPLC. The result can provide additional insight for the mixing state of aerosol particles which is still unknown in the atmospheric community. However, some results are not convincing and need more explanation.

1. Title should describe the topic and key words of the paper. 'Dry versus wet' is too simple. Also 'Implications' is not a main content of this paper. Please reconsider the title.

2. The authors selected glucose-NaCl mixtures as a model system consisting of a core-shell morphology. I am confusing whether the mixture really showed a core-shell morphology. Previous studies have established that particles containing organic/inorganic

salts underwent phase separation when the O:C of the organic material was smaller than ∼0.8 (Bertram et al., 2011; Krieger et al., 2012; Song et al., 2012; You et al., 2013; You et al., 2014). Since the O:C of glucose is 1.0, I expect the particle is present in one phase before NaCl effloresces. Once the particle reaches low relative humidity, it would show that nucleation of NaCl initiates effloresce process over the particle. The authors should state the discrepancy in the previous work in detail.

Bertram, A. K., Martin, S. T., Hanna, S. J., Smith, M. L., Bodsworth, A., Chen, Q., Kuwata, M., Liu, A., You, Y., and Zorn, S. R.: Predicting the relative humidities of liquid-liquid phase separation, efflorescence, and deliquescence of mixed particles of ammonium sulfate, organic material, and water using the organic-to-sulfate mass ratio of the particle and the oxygen-to-carbon elemental ratio of the organic component, Atmos. Chem. Phys., 11, 10995-11006, DOI 10.5194/acp-11-10995-2011, 2011.

Krieger, U. K., Marcolli, C., and Reid, J. P.: Exploring the complexity of aerosol particle properties and processes using single particle techniques, Chem. Soc. Rev., 41, 6631-6662, 10.1039/c2cs35082c, 2012.

Song, M., Marcolli, C., Krieger, U. K., Zuend, A., and Peter, T.: Liquid-liquid phase separation in aerosol particles: Dependence on O:C, organic functionalities, and compositional complexity, Geophys. Res. Lett., 39, Artn L19801, Doi 10.1029/2012gl052807, 2012.

You, Y., Renbaum-Wolff, L., and Bertram, A. K.: Liquid-liquid phase separation in particles containing organics mixed with ammonium sulfate, ammonium bisulfate, ammonium nitrate or sodium chloride, Atmos. Chem. Phys., 13, 11723-11734, 10.5194/acp-13-11723-2013, 2013.

You, Y., Smith, M. L., Song, M. J., Martin, S. T., and Bertram, A. K.: Liquid-liquid phase separation in atmospherically relevant particles consisting of organic species and inorganic salts, Int. Rev. Phys. Chem., 33, 43-77, 10.1080/0144235X.2014.890786, 2014.

[Figure]

3. Section 2.3. Please add information of the capacity for the size resolution of the AFM.

4. Experimental (Sect. 2.3, line 10) and results: Ten minutes would be too short to reach equilibrium with the surrounding air at the given RH of 25 – 30 %. Please check and compare with the equilibrium time from Grayson et al., 2017.

Grayson, J. W., Evoy, E., Song, M., Chu, Y. X., Maclean, A., Nguyen, A., Upshur, M. A., Ebrahimi, M., Chan, C. K., Geiger, F. M., Thomson, R. J., and Bertram, A. K.: The effect of hydroxyl functional groups and molar mass on the viscosity of non-crystalline organic and organic-water particles, Atmos. Chem. Phys., 17, 8509-8524, 10.5194/acp-17-8509-2017, 2017

5. Experimental and results: If the particles undergo RH decreasing gradually from high RH (i.e. ∼100%) to ∼25-30%, would you still expect core-sell morphology at the RH? It would be worth to try additional experiments.

---

## Author Comment (AC1) · 8 Jan 2019

***Reviewer1:*** *This paper investigated dry and wet deposition effects on the generated and impacted aerosol particles' mixing states that display core-shell morphology. In general, the techniques are well-used and the manuscript is well written. However, some suggestions are worth noting.*

**Authors:** We thank the reviewer for their time and diligence. We carefully addressed all of the specific questions and concerns raised by the reviewer, which is shown below. This reviewer expresses genuine support of our work. The comments improved the quality of the manuscript.

***Reviewer1:1*** *The abstract section is too long. Only key information is needed.*

**Authors:** Thanks for the comment; we revised the abstract to make it shorter. It is now 306 words in length. New abstract reads as follows,

*"Understanding the impact of sea spray aerosols (SSA) on the climate and atmosphere requires quantitative knowledge of their chemical composition and mixing states. Furthermore, single particle measurements are needed to accurately represent large particle-to-particle variability. To quantify the mixing state, organic volume fraction (OVF), defined as the relative organic volume with respect to the total particle volume, is measured after generating and collecting aerosol particles, often using deposition impactors. In this process, the aerosol streams are either dried or kept wet prior to impacting on solid substrates. However, the atmospheric community has yet to establish how dry versus wet aerosol deposition influences the impacted particle morphologies and mixing states. Here, we apply complementary offline single particle atomic force microscopy (AFM) and bulk ensemble high performance liquid chromatography (HPLC) techniques to assess the effects of dry and wet deposition modes on the substrate-deposited aerosol particles' mixing states. Glucose and NaCl binary mixtures that form core-shell particle morphologies were studied as model systems, and the mixing states were quantified by measuring the OVF of individual particles using AFM and compared to the ensemble measured by HPLC. Dry deposited single particle OVF data positively deviated from the bulk HPLC data by up to 60%, which was attributed to significant spreading of the NaCl core upon impaction with the solid substrate. This led to underestimation of the core volume. This problem was circumvented by a) performing wet deposition and thus bypassing the effects of the solid core spreading upon impaction and b) performing a hydration-dehydration cycle on dry deposited particles to restructure the deformed NaCl core. Both approaches produced single particle OVF values that converge well with the bulk and expected OVF values, validating the methodology. These findings illustrate the importance of awareness in how conventional particle deposition methods may significantly alter the impacted particle morphologies and their mixing states."*

***Reviewer1:2*** *As we know, levoglucosan dominates the sugar particulates, while little glucose exists in the atmosphere. Meanwhile, more details about glucose (or particle phase sugar) should be added in the Introduction section.*

**Authors:** Levoglucosan is a major component of biomass burning aerosol, however it is has not been detected in nascent sea spray aerosol, which is the chemical system that is modeled in this set of experiments. In regard to sea spray aerosol, glucose has been identified as the most abundant carbohydrate, contributing 5.2% and 14.4% of $PM_{2.5}$ and $PM_{10-2.5}$ (Jayarathne et al., 2016).

This is now reflected in the text, at the last paragraph of the 1 Introduction. Following sentences were included,

"Both chemical systems are highly relevant to the SSA, with glucose up to 5.2% and 14.4% of the total organic mass of $PM_{2.5}$ and $PM_{10-2.5}$ SSA, respectively. (Jayarathne et al., 2016) Further, both glucose and NaCl are surface inactive species, thus OVF is not expected to be size-dependent. (Cochran et al., 2017)"

*Reviewer1:3 Section 3.1: Authors employed the RH of 25% to measure OVF. However, RH in the real atmosphere is usually higher than 30%. Additionally, whether the authors considered the particle hygroscopicity?*

**Authors:** It is true that the RH in the atmosphere fluctuates. However, controlling of RH in our experiments, to quantify the organic volume fraction, is solely to minimize the amount of water within the particles to improve the accuracy of this methodology. As outlined in Section 3.1 Morphology of phase-separated binary component particles, we previously stressed that,

"Here, RH was maintained at a constant range of 25 – 35% throughout the imaging experiment, to minimize "phase bleeding" and water uptake (Fig. S1). In this work, phase bleeding describes an instance in which the viscosity of the two phase-separated materials is too similar at a given RH, and therefore the phase contrast between organic and inorganic components is relatively weak in the AFM images. A high phase contrast image aids in correctly differentiating the core and shell phase boundaries. For this particular system, a dramatic increase in phase bleeding is evident below 25% RH, due to a significant increase in the glucose viscosity as it becomes more solid-like, which closely resembles the NaCl viscosity at this RH value (Fig. S1). (Song et al., 2016) Since phase imaging inherently relies on measuring differences in tip-sample interactions originating from different viscoelastic properties, lowering the RH will further converge the two different viscosities of organic and inorganic components together, lessening the accuracy of the core and shell phase boundary determination. Although higher RH values would produce less "phase bleeding", significant particle water uptake will also introduce erroneous OVF values, due to an increase in particle volume from water uptake."

The authors did indeed consider the hygroscopicity of the chemical systems. In fact, the smooth and slow uptake of water, unique for organic species such as saccharides, contributes to access to solid, semisolid, and liquid phase states for glucose, which was mentioned in the introduction. Under Section 3.1 Morphology of phase-separated binary component particles, the following sentence was revised to address the influence of the change in organic shell volume due to hygroscopic growth (only 3% here for glucose):

"The value of 25% RH was thus used for the measurements of OVF, because growth in the organic shell size due to water uptake was measured to be less than 1.03, or merely 3% growth while showing excellent phase contrast. (Lee et al., 2017a)"

*Reviewer1:4 For the equation S1, which kind of density is used here? Please explain in detail.*

**Authors:** Thanks for the comment. The solute densities of pure glucose and NaCl were used. The Ensemble average OVF comparison between AFM and bulk in the supporting information now reads,

"In this case, the concentrations of glucose, sodium and chloride were determined in aqueous extracts of filter samples and were field blank subtracted. Masses of these species were converted to volumes, and used to calculate $OVF_{bulk}$ following Equation S1,

$$OVF_{bulk} = \frac{\dfrac{m_{org}}{\rho_{org}}}{\dfrac{m_{org}}{\rho_{org}} + \dfrac{m_{inorg}}{\rho_{inorg}}} \qquad \text{Eq. S1}$$

where $OVF_{bulk}$ is organic volume fraction of bulk, $m_{org}$ & $m_{inorg}$ are masses, and $\rho_{org}$ & $\rho_{inorg}$ are pure solute densities of organic and inorganic components, respectively.

***Reviewer1:5*** *The authors used one paragraph to describe the t-test and null hypothesis. But t-test is a simple statistical tool and only the significant result should be put in manuscript.*

**Authors:** Thanks for the comment. We moved the explanation of the t-test and null hypothesis in Sections 3.2 to the supporting information. In addition, replacements in sections 3.2 and 3.4 now read,

"OVF measured from dry deposited particles and wet-deposited particles were compared with a two-tailed t-test, which demonstrated that these two datasets are significantly different ($p < 0.0001$, see SI)."

"To confirm this statistically, same student's t test methodology was employed, which demonstrated that these two datasets are significantly different ($p < 0.0001$, see SI)."

***Reviewer1:6*** *Conclusion section. The atmospheric implications of the experimental results should be discussed.*

**Authors:** To begin, this work heavily implicates that the particle generation method may affect measuring of organic volume fraction using offline, single particle techniques. This may lead to erroneous representation of the mixing state, which may affect the accuracy of climate models that require this knowledge to understand aerosol-cloud interactions.(Ault and Axson, 2017) This reference was added into the introduction. On the first paragraph of the 1 Introduction, we added,

"Uncertainty in the mixing state, even with the knowledge of the chemical composition, may produce erroneous predictions of cloud activation from individual particles. (Ault and Axson, 2017)"

Moreover, the following was added to the conclusion,

"Overall, our findings provide implications of aerosol generation on accurately identifying the mixing state of individual, phase separated particles, important for the atmospheric community and climate predicting models that heavily rely on not only the knowledge of the chemical composition, but also the resultant mixing states of atmospherically-relevant aerosol particles."

---

## Author Comment (AC2) · 8 Jan 2019

**Reviewer2:** The authors describe the effect of dry and wet deposition on the mixing states of organic/NaCl particles using atomic force microscopy and bulk ensemble HPLC. The result can provide additional insight for the mixing state of aerosol particles which is still unknown in the atmospheric community. However, some results are not convincing and need more explanation.

**Authors:** We thank the reviewer for their time and diligence. We carefully addressed all of the specific questions and concerns raised by the reviewer, which is shown below. This reviewer expresses genuine support of our work. The comments improved the quality of the manuscript.

**Reviewer2:** 1. Title should describe the topic and key words of the paper. 'Dry versus wet' is too simple. Also 'Implications' is not a main content of this paper. Please reconsider the title.

**Authors:** Thanks for the comment. The implications relate the dry or wet nature of the particle deposition to the measured organic volume fraction. In consideration of this, we changed the title to the following,

"Effect of dry or wet substrate deposition on aerosol organic volume fraction"

**Reviewer2:** 2. The authors selected glucose-NaCl mixtures as a model system consisting of a coreshell morphology. I am confusing whether the mixture really showed a core-shell morphology. Previous studies have established that particles containing organic/inorganic salts underwent phase separation when the O:C of the organic material was smaller than 0.8 (Bertram et al., 2011; Krieger et al., 2012; Song et al., 2012; You et al., 2013; You et al., 2014). Since the O: C of glucose is 1.0, I expect the particle is present in one phase before NaCl effloresces. Once the particle reaches low relative humidity, it would show that nucleation of NaCl initiates effloresce process over the particle. The authors should state the discrepancy in the previous work in detail.

Bertram, A. K., Martin, S. T., Hanna, S. J., Smith, M. L., Bodsworth, A., Chen, Q., Kuwata, M., Liu, A., You, Y., and Zorn, S. R.: Predicting the relative humidities of liquid-liquid phase separation, efflorescence, and deliquescence of mixed particles of ammonium sulfate, organic material, and water using the organic-to-sulfate mass ratio of the particle and the oxygen-to-carbon elemental ratio of the organic component, Atmos. Chem. Phys., 11, 10995-11006, DOI 10.5194/acp-11-10995-2011, 2011.

Krieger, U. K., Marcolli, C., and Reid, J. P.: Exploring the complexity of aerosol particle properties and processes using single particle techniques, Chem. Soc. Rev., 41, 6631- 6662, 10.1039/c2cs35082c, 2012.

Song, M., Marcolli, C., Krieger, U. K., Zuend, A., and Peter, T.: Liquid-liquid phase separation in aerosol particles: Dependence on O: C, organic functionalities, and compositional complexity, Geophys. Res. Lett., 39, Artn L19801, Doi 10.1029/2012gl052807, 2012.

You, Y., Renbaum-Wolff, L., and Bertram, A. K.: Liquid-liquid phase separation in particles containing organics mixed with ammonium sulfate, ammonium bisulfate, ammonium nitrate or sodium chloride, Atmos. Chem. Phys., 13, 11723-11734, 10.5194/acp- 13-11723-2013, 2013.

You, Y., Smith, M. L., Song, M. J., Martin, S. T., and Bertram, A. K.: Liquid-liquid phase separation in atmospherically relevant particles consisting of organic species and inorganic salts, Int. Rev. Phys. Chem., 33, 43-77, 10.1080/0144235X.2014.890786, 2014.

**Authors:** This is an interesting comment. To begin, the phase separation/ core shell morphology we observe is solid-semisolid phase separation, not liquid-liquid phase separation. This is evident in Figure 1,

where phase images clearly identify the solid core surrounded within a shell. This solid NaCl is also evident in Figure 3. Moreover, we are not aware of AFM phase imaging capable of identifying liquid-liquid phase separation.

The reviewer's comment that the glucose + NaCl mixture, above the deliquescence point is likely not liquid-liquid phase separated, is reasonable. In this work, we do not claim that this methodology applies to the glucose + NaCl mixture beyond the deliquescence point at relative humidity greater than 75%. Below the efflorescence point, however, the NaCl would indeed become a solid, which is again, evident in Figures 1 and 3. Therefore, we do not see any discrepancy to note, as the cited works pertain to different type of phase separation.

That being said, publications cited by this reviewer are credible works that studied liquid-liquid phase separation, and any that were not previously cited, are now cited within this manuscript. To the last paragraph of the introduction, we added,

"Unlike liquid-liquid phase separation, to the best of our knowledge, parameterization to predict a solid-semisolid or solid-liquid phase separation does not yet exist.(Bertram et al., 2011;You et al., 2013;You et al., 2014;Krieger et al., 2012;Song et al., 2012)"

**Reviewer2:** 3. Section 2.3. Please add information of the capacity for the size resolution of the AFM.

**Authors:** Good point. We added in Section 2.3 AFM imaging,

"The microscope permits (sub)nanometer-level spatial resolution and 1 pN force resolution. (Binnig et al., 1986;Santos et al., 2011;Gan, 2009;Gerber, 2017)"

**Reviewer2:** 4. Experimental (Sect. 2.3, line 10) and results: Ten minutes would be too short to reach equilibrium with the surrounding air at the given RH of 25 – 30 %. Please check and compare with the equilibrium time from Grayson et al., 2017.

Grayson, J. W., Evoy, E., Song, M., Chu, Y. X., Maclean, A., Nguyen, A., Upshur, M. A., Ebrahimi, M., Chan, C. K., Geiger, F. M., Thomson, R. J., and Bertram, A. K.: The effect of hydroxyl functional groups and molar mass on the viscosity of noncrystalline organic and organic-water particles, Atmos. Chem. Phys., 17, 8509-8524, 10.5194/acp-17-8509-2017, 2017

**Authors:** Thanks for the comment. Time required to equilibrate with respect to the relative humidity of the surrounding depends on the rate of diffusion. In Grayson et al., particles of size range of 55 – 80 um in diameter are probed at 19% relative humidity. This is shown to correspond to characteristic time of mass-transport and mixing of water to be between 365 to 771 minutes.

However, the largest particles we probed are at best, 0.9 um in volume equivalent diameter, as illustrated by Figures 2 and 4. Since the diffusion coefficient is inversely proportional to diameter, one can expect a proportional decrease in the equilibration time. For example, decrease in size from 80 to 0.8 um is approximately 2 orders of magnitude, so equilibration time of approximately 7.7 minutes is expected for 0.8 um sized particles.

This is further supported by previous works that studied the half time of particle radius as a function of RH, where a 1.6 um diameter sucrose at 20% RH showed half time of approximately 250 seconds. Even

the largest particle that we studied, would be a factor of nearly 2 lower in diameter, and thus we expect the half time to further significantly decrease from 250 seconds. (Lu et al., 2014)

This is now reflected in the manuscript, in Section 2.3 AFM imaging,

"This timeframe is reasonable given the diffusive nature of water transport within a particle, which is dependent on the particle size.(Grayson et al., 2017;Lu et al., 2014)"

**Reviewer2:** 5. Experimental and results: If the particles undergo RH decreasing gradually from high RH (i.e. 100%) to 25-30%, would you still expect core-sell morphology at the RH? It would be worth to try additional experiments.

**Authors:** For this mixture, we would indeed expect the core-shell morphology. In fact, in our hydration-dehydration cycles, the RH is raised up to 80%, which is beyond the deliquescence point. Further increasing the RH to nearly 100% and then lowering down to 25-30% would not affect our results, as the difference will lie in the amount of water that the particle uptakes at high RH. Since the core-shell morphology is also reproducible with multiple hydration-dehydration cycles leading to similar OVF, we feel there is no obvious need to do any additional experiments.